# DNA Barcoding and Fertilization Strategies in *Sideritis syriaca* subsp. *syriaca*, a Local Endemic Plant of Crete with High Medicinal Value

**DOI:** 10.3390/ijms25031891

**Published:** 2024-02-04

**Authors:** Konstantinos Paschalidis, Dimitrios Fanourakis, Georgios Tsaniklidis, Ioannis Tsichlas, Vasileios A. Tzanakakis, Fotis Bilias, Eftihia Samara, Ioannis Ipsilantis, Katerina Grigoriadou, Ioulietta Samartza, Theodora Matsi, Georgios Tsoktouridis, Nikos Krigas

**Affiliations:** 1Department of Agriculture, School of Agricultural Sciences, Hellenic Mediterranean University, 71410 Heraklion, Greece; dfanourakis@hmu.gr (D.F.); giannistsixlas@gmail.com (I.T.); vtzanakakis@hmu.gr (V.A.T.); 2Hellenic Agricultural Organization (ELGO-DIMITRA), Institute of Olive Tree, Subtropical Crops and Viticulture, 73134 Chania, Greece; tsaniklidis@elgo.iosv.gr; 3Soil Science Laboratory, School of Agriculture, Aristotle University of Thessaloniki, 54124 Thessaloniki, Greece; fbilias@agro.auth.gr (F.B.); iipsi@agro.auth.gr (I.I.); thmatsi@agro.auth.gr (T.M.); 4Institute of Plant Breeding and Genetic Resources, Hellenic Agricultural Organization Demeter, 57001 Thessaloniki, Greece; katgrigoriadou@elgo.gr (K.G.); isamartza@gmail.com (I.S.); nkrigas@elgo.gr (N.K.); 5Theofrastos Fertilizers, Industrial Area of Korinthos, Irinis & Filias, Ikismos Arion, Examilia, 20100 Korinthos, Greece

**Keywords:** molecular markers, genetic characterization, antioxidant, carotenoids, flavonoids, Greece, phenols, biostimulant, malotira, mountain tea

## Abstract

Herein, we applied DNA barcoding for the genetic characterization of *Sideritis syriaca* subsp. *syriaca* (Lamiaceae; threatened local Cretan endemic plant) using seven molecular markers of cpDNA. Five fertilization schemes were evaluated comparatively in a pilot cultivation in Crete. Conventional inorganic fertilizers (ChFs), integrated nutrient management (INM) fertilizers, and two biostimulants were utilized (foliar and soil application). Plant growth, leaf chlorophyll fluorescence, and color were assessed and leaf content of chlorophyll, key antioxidants (carotenoids, flavonoids, phenols), and nutrients were evaluated. Fertilization schemes induced distinct differences in leaf shape, altering quality characteristics. INM-foliar and ChF-soil application promoted yield, without affecting tissue water content or biomass partitioning to inflorescences. ChF-foliar application was the most stimulatory treatment when the primary target was enhanced antioxidant contents while INM-biostimulant was the least effective one. However, when the primary target is yield, INM, especially by foliar application, and ChF, by soil application, ought to be employed. New DNA sequence datasets for the plastid regions of *pet*B/*pet*D, *rpo*C1, *psb*K-*psb*I, and *atp*F/*atp*H were deposited in the GenBank for *S. syriaca* subsp. *syriaca* while the molecular markers *rbc*L, *trn*L/*trn*F, and *psb*A/*trn*H were compared to those of another 15 *Sideritis* species retrieved from the GenBank, constructing a phylogenetic tree to show their genetic relatedness.

## 1. Introduction

Genomic approaches supported by molecular biology and bioinformatics are of critical importance to document and conserve biodiversity [1]. DNA sequences used as “barcodes” to define genetically biological entities (species) represent a fast, low-cost, reliable, and uncomplicated solution for plant species identification, further allowing insight into their phylogenetic relations [2,3]. In various terrestrial medicinal-aromatic plants (MAPs), molecular barcoding has been widely applied, using short regions of nuclear or chloroplast DNA to genetically characterize and identify taxa at the species or even at the subspecies level [4], including, among others, the *trn*L, *trn*T-*trn*L, *psb*A-*trn*H, and ITS molecular markers. Previous studies on members of Lamiaceae (a major family of MAPs worldwide) have used various molecular markers and DNA tools for the successful identification of many members belonging to different genera of Lamiaceae, with some genera receiving more attention than others due to complex taxonomical issues, such as members of the genus *Sideritis* [5,6,7,8,9,10,11,12,13,14,15,16,17]. Specimens of *Sideritis* spp. are often difficult to identify solely on the basis of morphological features as the latter may overlap between different wild-growing species or they are occasionally obscured due to natural hybridization, resulting in intermediate characteristics [9]; in addition, and when in close proximity within conservation facilities, different ex situ cultivated taxa (species and subspecies) that have been originally collected from wild-growing populations may finally result in hybridized individuals due to uncontrolled pollination syndromes (open pollination).

The first molecular DNA study for the *Sideritis* taxa was initiated in 1999 using RAPDs for fingerprinting fifteen plant populations of *S. pusilla* (Lange) Pau and their putative parental species *S. hirsuta* L. and *S. leucantha* Cav. [5], shedding light on taxonomic affinities on the basis of morphological, biogeographical, and genetic data with comparative analysis of variance within and between populations; meanwhile, the first DNA sequences submitted in GenBank were back in 2002 for the *trn*L gene of cpDNA for five different *Sideritis* species [6]. Another significant phylogenetic study has analyzed GenBank datasets focused on 23 Macaronesian *Sideritis* taxa (species and subspecies) using the ITS, *trn*L, and *trn*T-*trn*L molecular markers to consolidate the genetic relatedness, hybridization, and evolutionary affinities among them [9,11]. More recently, another 93 specimens of the Greek *Sideritis* taxa have been deposited for the ITS, *psb*A-*trn*H (PopSet: 972386283), *rbc*L (PopSet: 972385939), and *mat*K (PopSet: 972386103) molecular markers [18]; however, there is no relevant study published to date. The nuclear molecular marker ITS has been broadly used in such studies [6,14,15,17]; whereas, multiple regions from the cpDNA have also been used for species identification [6,10,11,12,13,16]. Many fingerprinting studies to date have employed genetic analyses using Amplified Fragment Length Polymorphisms (AFLPs) for several subpopulations of the local Greek endemic *S. euboea* Heldr. [8], Random Amplified Polymorphic DNA (RAPD) markers, [19,20] and 12 URP to investigate the inter-population genetic diversity of *S. raeseri* Boiss. and Heldr. [7].

Species-wise, this investigation was focused on a perennial local endemic plant of Crete (Greece), namely, *S. syriaca* L. subsp. *syriaca* (Lamiaceae), commonly called in Crete ‘malotira’ (or Cretan mountain tea). This taxon is wild-growing on rocky substrates at high altitudes (1000–2200 m) of the major mountain massifs of Crete and flowers in June and July [21]. In higher altitudes, the plants enter at flowering stage later [22]. To date, *S. syriaca* subsp. *syriaca* may also be found as a cultivated crop and its dry product yield reaches 1000–1500 kg of dry weight per hectare [22]. It is resistant to drought, having low nutrient needs, and the perfect soil pH is between 6.9 and 8 [22]. Malotira represents a contemporary yet traditional herbal medicinal product, which is widely used for tea preparation and as a food ingredient with well-documented therapeutic indications [23] approved by the European Medicines Agency (https://www.ema.europa.eu/en/medicines/herbal/sideritis-herba, accessed 30 November 2023). Although it has been recently cultivated at a small scale in the Omalos plateau (Chania, Crete) as a valuable locally native MAP [24], it is threatened with extinction [25] due to the over-harvesting of sizeable volumes of plant material directly sourced from wild-growing populations of Lefka Ori in Chania, Crete (https://flashnews.gr/post/694054/ena-ntokimanter-gia-tin-kritiki-malotira-eftase-mechri-ti-germania-vinteo/, accessed 30 November 2023); such harvested (or cultivated at lesser extent) material is traded in local markets of Crete and is also exported as a local MAP product.

In many countries, traditional and complementary medicines are still a major health resource, with herbal materials used for therapeutic purposes [26,27]. In China alone, ca. 40% of the administered medicines are of herbal origin [28]. The latter is also coupled with an emerging global trend of using MAPs and natural products in Western societies [29]. Notably, several industrially produced (pharmaceutical) drugs are still based on MAPs [27] and the market and public demands for a wide range of plant-derived substances (e.g., fragrances, cosmetics, color/flavor additives, etc.) continue to rise [28,30,31]. In this context, the growing request for MAPs, however, inevitably imperils their wild-growing populations [28,32], like those of the threatened focal taxon in this study. It is known that unmonitored or poorly controlled trade, coupled with unsuitable harvesting methods, and over-exploitation pressure on wild-growing populations may imperil several range-restricted species, such as the local endemic *S. syriaca* subsp. *syriaca* [17], as well as maybe jeopardizing the ecological equilibrium at the habitat/ecosystem level [32]. Nonetheless, introducing focal species of conservation concern and economic interest in agricultural settings, such as the focal taxon herein, can satisfy the growing demand for specific MAPs without further depleting wild-growing phytogenetic resources. To this end, applied research is needed, such as the documentation of origin and consolidated taxonomic identity of plant material used, species-specific propagation protocols, cultivation guidelines, and fertilization schemes, thus facilitating conservation actions and enabling sustainable exploitation in different economic sectors [24].

Cultivation-wise, any plant demand for nutrients is often conventionally met by inorganic fertilizers. Accumulative evidence, however, suggests that the over-application or inefficient utilization of these fertilizers are major environmental pollution drivers with cumulative alarms [33]. Typical examples include the degradation of soil quality and nutrient loading of water bodies with deleterious effects on aquatic ecosystems [33,34]. A more eco-friendly and more sustainable practice of satisfying plant nutrient needs is using organic fertilizers, which produce a rather limited environmental impact compared to conventional ones [35]. Fertilization management can be further improved by applying biostimulants [36], which may increase root nutrient acquisition, enhancing, in this way, fertilization efficiency [36]. In this perspective, the joint addition of conventional inorganic fertilizers, organic fertilizers, and biostimulants (collectively termed integrated nutrient management; INM) can potentially ensure plant growth and productivity at a lower environmental cost [37,38]. Such applied research lines have also been suggested as indispensable stepping stones for the sustainable exploitation and pilot cultivation of neglected and underutilized local Cretan endemic plants with potential economic interest [25].

Market-wise, the herbal material quality is initially shaped by the fertilization scheme applied [35] and includes a wide range of features, the relative importance of which is determined by the intended use [31]. Leaf-soluble sugars affect all plant life cycle stages by playing a role in the plant’s energy balance and interacting with signaling molecules [39]. The level of antioxidant substances (often referred to as antioxidants) is commonly an important index of plant material quality [34]. Antioxidants are widely recognized as potential health promoters and they have well-documented preventative and therapeutic effects against several chronic diseases [40]. These include carotenoids, flavonoids, and phenolics, which are characterized by powerful antioxidant ability [34], playing a title role in the dealing with and avoidance of most diseases [41]; the herein focal taxon has also rich content in such beneficial compounds [24]. It has been suggested that organic fertilizers and integrated nutrient management (INM) strategies may stimulate the herbal material antioxidant profile more than conventional inorganic fertilizers [40,42,43,44,45,46]. Furthermore, as these strategies help in avoiding societal charges of nitrogen losses, the paybacks of their use might considerably shade their expenses, rendering them extremely highly suggested for incorporation in fertilization schemes [47]. Such fertilization schemes have also been associated with enhanced chlorophyll content [48]. Chlorophyll content determines leaf greenness, which is another measure of herbal material quality, being traditionally employed by processors, distributors, and consumers [31] so it is often used for the monitoring of entire plant health [49].

Context-wise, the present investigation aimed to genetically characterize the local endemic *S. syriaca* subsp. *syriaca* (malotira or Cretan mountain tea) using seven plastid molecular markers, four of which were employed for the first time in members of the genus *Sideritis*. The molecular markers selected as suitable plant DNA barcodes are widely used, are suggested by several studies, and have been routinely tested in terms of amplification and sequencing success (universality), intra-/inter-specific variation, and resolving power [3,50,51]. These new DNA sequences can be used for reliable taxonomic identification and traceability of commercial products of *Sideritis herba* [24], also supporting the genetic authentication and identity consolidation of related commercial products. Another aim of this study, in response to the absence of species-specific data, was to comparatively investigate the effects of different fertilization treatments and application methods in a pilot field cultivation of *S. syriaca* subsp. *syriaca* in Heraklion, Crete; the latter was used to define a species-specific fertilization protocol able to promote plant growth and/or key herbal quality features (i.e., color intensity, antioxidant metabolite content) for sustainable and commercial exploitation.

## 2. Results

### 2.1. Molecular Characterization of the Studied Sideritis syriaca *subsp.* syriaca

The DNA sequences of *S. syriaca* subsp. *syriaca* GR-1-BBGK-15,5939 generated in the present study for the molecular markers *pet*B/*pet*D (999 bp), *rbc*L (718 bp), *trn*L/*trn*F (695 bp), *rpo*C1 (476 bp), *psb*A/*trn*H (534 bp), *psb*K/*psb*I (329 bp), and *atp*F/*atp*H (498 bp) were deposited in the Genbank obtaining the accession numbers OR909054-OR909060.

A unified phylogenetic tree was constructed for the molecular markers *trn*L/*trn*F and *rbc*L for a total of 15 Sideritis taxa (Figure 1). The concatenated sequence used for this phylogenetic tree had a length of 1233 bp; within this sequence, a total of 43 single nucleotide polymorphisms (SNPs), a polyT (9 Ts) region, and an indel (insertion of TGAA for *S. romana* L.) have been identified (Appendix A). In the phylogenetic tree, *S. syriaca* subsp. *syriaca* was clearly separated from closely related species, such as *S. euboea* and *S. scardica* (Figure 1).

The DNA sequence of the present study (OP909056) for *rbc*L was identical (100%) to *S. syriaca* subsp. *syriaca* (KT315757) and identical (100%) to another 19 specimens corresponding to *S. raeseri* subsp. *raeseri*, *S. euboea*, and *S. scardica* Griseb. that originated from Greece; it was also similar (99.86%) to *S. euboea* AF501999 from Mt. Dirphys, which was wrongly labeled as *S. syriaca*, as well as to *S. hyssopifolia* L. (AF501995) (Appendix A). The DNA sequence OP909054 of the present study for *psb*A-*trn*H was identical (100% similarity) to the respective sequences of *S. syriaca* subsp. *syriaca* deposited in the GenBank (KT633327, KT633328, KT633350) [18]. Surprisingly, there were nucleotide differences among 90 more *Sideritis* taxa that originated from Greece, with results shown to be highly problematic (Appendix A).

Basic Local Alignment Search Tool (BLAST) results showed that the *pet*B/*pet*D molecular marker exhibited a 98.60% similarity with *Stenogyne haliakalae* Wawra (NC_029817). The sequence generated for the *trn*L/*trn*F region in this study matched with *S. scardica* (AF335658) by 99.81%; whereas, the *rpo*C1 sequence matched to *Stachys byzantina* K.Koch (NC_029825) by 99.37%. The sequence for the *psb*K/*psb*I intergenic spacer matched with *Anisomeles indica* (L.) Kuntze (NC_046781) and *Leucosceptrum canum* Sm. (NC_051966) by 94.84%. Finally, the sequence for *atp*F/*atp*H matched *Stachys byzantina* (NC_029825) by 98.20% [18].

### 2.2. Fertilization Scheme Exerted Minor Effects on Leaf Color

At all three growth stages, the leaf SPAD value was not affected by the fertilization treatment (Figure 2). At the vegetative stage, plants receiving INM-fa had a higher I_AD_ value (i.e., being less green) from all treatments other than INM-sa (Appendix A). At the early flowering stage, plants receiving ChF-sa showed higher I_AD_ values than plants treated with ChF-fa and MPE-sa (Appendix A). At the full flowering stage, plants receiving ChF-sa had a higher I_AD_ value as compared to plants treated with ChF-fa, INM-sa, or MPE-sa (Appendix A). However, at the latter two growth stages, all fertilization treatments were similar to the control (Appendix A).

At all three growth stages, the leaf L value was not affected by the fertilization scheme (Appendix A). At the vegetative stage, plants receiving ChF-sa had a lower a* value as compared to control plants or plants receiving ChF-fa or INM-sa (Appendix A). At the early flowering stage, plants receiving INM-sa or ChF-sa showed lower a* values than control plants or plants treated with ChF-fa (Appendix A). At the full flowering stage, plants receiving INM-sa had a lower a* value than plants receiving INM-fa or ChF-fa, though all fertilization treatments were similar to the control (Appendix A).

At the early flowering stage, plants receiving INM-sa had a higher b* value as compared to control plants. However, the b* value of the NM-sa treatment was similar to that of the rest fertilization treatments, which were also similar to the control (Appendix A). At vegetative and full flowering stages, the fertilization treatment did not affect leaf b* value (Appendix A).

### 2.3. Fertilization Scheme Induced Limited Effects on Leaf Photosynthetic Performance

At three growth stages, the effect of fertilization treatment on overall photosynthetic efficiency was assessed by in situ F_v_/F_m_ measurements (Figure 3). At the vegetative stage, plants receiving INM-fa had the highest F_v_/F_m_ value (Figure 3A). At the early flowering stage, plants receiving INM-sa had higher F_v_/F_m_ values as compared to control plants (Figure 3B). At the full flowering stage, the leaf F_v_/F_m_ value was not affected by the fertilization scheme (Figure 3C).

### 2.4. Fertilization Scheme Induced Distinct Leaf Shape Profiles

To determine the effect of the fertilization scheme on leaf form, four shape indicators (aspect ratio, circularity, roundness, solidity) were employed. Control plants showed a higher aspect ratio as compared to plants receiving ChF-sa or MPE-sa (Figure 4A). The lowest circularity was noted in control plants (Figure 4B). Plants receiving MPE-sa had the highest circularity from all treatments, other than ChF-sa (Figure 4B). Plants under MPE-sa had higher roundness than control plants (Figure 4C). The lowest solidity was noted in control plants (Figure 4D). Plants treated with MPE-sa had the highest solidity of all treatments, other than plants receiving INM-sa (Figure 4D).

### 2.5. INM and ChF-sa Stimulated Plant Growth, without Affecting Biomass Allocation to Generative Organs

INM with either application method (i.e., INM-fa or INM-sa) and ChF-sa improved biomass accumulation as compared to the remaining treatments, including the control (Figure 5A). These differences in above-ground dry mass were not associated with variation in tissue relative water content (Figure 5B). Plants receiving INM-sa had higher partitioning to generative organs as compared to plants receiving ChF-fa; although, all fertilization treatments were similar to the control (Figure 5C).

### 2.6. Fertilization Scheme Affected Leaf Chlorophyll Content

Plants receiving foliar fertilization (i.e., INM-fa, ChF-fa) or INM-sa had higher leaf chlorophyll content as compared to the control (Figure 6A). Plants receiving ChF-sa or MPE-sa had lower leaf chlorophyll content as compared to the control while the latter had the lowest content (Figure 6A).

### 2.7. Fertilization Scheme Affected Leaf Antioxidant Compound Content

Carotenoids, flavonoids, and phenols are critical nonenzymatic antioxidants. Plants receiving foliar fertilization (i.e., INM-fa, ChF-fa) or INM-sa had higher leaf carotenoid content as compared to the control; whereas, plants receiving MPE-sa had lower leaf carotenoid content as compared to the control (Figure 6B).

The highest leaf total phenolic content was noted in plants receiving ChF-fa, followed by plants treated with INM-sa or ChF-sa (Figure 6C). Plants treated with INM-sa or ChF-sa had higher total phenolic content as compared to the control; whereas, the ones receiving MPE-sa had the lowest total phenolic content (Figure 6C).

Plants receiving INM-fa, INM-sa, or ChF-sa had lower leaf flavonoid content as compared to the control (Figure 6D). Again, the lowest leaf flavonoid content was noted in plants receiving MPE-sa (Figure 6D).

### 2.8. Fertilization Scheme Affected the Soluble Sugar Content of Leaves

Plants receiving ChF-fa had higher leaf-soluble sugar content than the control; whereas, the ones receiving INM-sa or MPE-sa had lower leaf-soluble sugar contents (Figure 7).

### 2.9. Leaf and Inflorescence Nutrient Analysis

Among all treatments (including the control), the highest N concentration in leaves was observed when plants were cultivated under integrated nutrient management fertilization applied to soil (INM-sa), followed by that of plants under INM-fa. Conversely, the highest K concentration was noted in plants under foliar application of conventional inorganic fertilizers (ChF-fa), followed by that of plants under ChF-sa. As far as P is concerned, this aforementioned trend was observed for plants receiving ChF-sa or INM-fa. Contrary to N-P-K, no significant differences were observed for Ca or Mg concentrations with any of the applied treatments (Table 1).

On the other hand, the results regarding the leaf concentration of micro-nutrients in *S. syriaca* subsp. *syriaca* were rather inconclusive. Specifically, no treatment effect was evidenced for Zn and the same stands for Cu, where, although ANOVA results indicated significant differences among treatments, none of them significantly differed from the control. On the contrary, the integrated nutrient management (INM) fertilization along with the biostimulant treatment (MPE) applied through soil application exhibited the highest concentrations for Fe while the conventional fertilization by foliar application (ChF-fa) proved to be more effective than all other treatments in supplying the highest Mn leaf concentration (Table 2).

Moreover, ANOVA of the floral nutrient data showed certain differences among treatments (Table 3 and Table 4). The highest N content was observed with INM-sa; whereas, regarding P, the highest contents were found with INM-fa and ChF-sa (Table 3).

Regarding the inflorescence micro-nutrient content of *Sideritis syriaca* subsp. *syriaca* (Table 4), the observed results were rather neutral regarding the beneficial effect of fertilization treatment on the particular plant parameter (Table 4).

## 3. Discussion

### 3.1. Molecular Characterization of the Studied Sideritis syriaca *subsp.* syriaca

No data were available in the GenBank for the *Sideritis* taxa for the molecular markers *pet*B/*pet*D, *rpoC1*, *psb*K/*psb*I, and *atp*F/*atp*H; thus, such data are first-time furnished herein for *S. syriaca* subsp. *syriaca*. However, there were 161 entries available on the Genbank for the *trn*L/*trn*F molecular marker corresponding to 49 *Sideritis* taxa, 96 entries belonging to 8 *Sideritis* taxa for *psb*A/*trn*H, and 45 entries of 22 *Sideritis* taxa for *rbc*L [18]. The genetic identification of plant species combining taxonomic identification and modern molecular techniques based on DNA sequences has the power to effectively characterize both morphologically and genetically given specimens [53]. To date, DNA barcoding has been widely used as a powerful tool in combination with bioinformatic analysis in ecological, taxonomic, comparative phylogenetic, and biodiversity conservation studies across various taxonomic levels [3], including the distinction of members of the Lamiaceae family [15,54] and, more specifically, of the genus *Sideritis* [11,13,14].

DNA barcoding was proved to be a valid technique for the discrimination of the herein-studied *S. syriaca* subsp. *syriaca* genotype, further enhancing the classical taxonomic identification based on morphological features and offering insight into the phylogenetic relationships of closely related taxa. In the frame of this study, seven new sequences (i.e., *pet*B/*pet*D, *rbc*L, *trn*L/*trn*F, *rpo*C1, *psb*A/*trn*H, *psb*K-*psb*I, and *atp*F/*atp*H) were generated and deposited in the GenBank for *S. syriaca* subsp. *syriaca* GR-1-BBGK-15,5939, obtaining unique accession numbers and creating genetic documentation for the focal taxon, while four DNA sequences of the plastid molecular markers *pet*B/*pet*D, *rpo*C1, *psb*K-*psb*I, and *atp*F/*atp*H have been reported and deposited to the GenBank for the first time. The phylogenetic analysis with the molecular markers *rbc*L and *trn*L/*trn*F confirmed the identity of the specimen and indicated its phylogenetic position compared to other *Sideritis* taxa (Figure 1). *S. syriaca* subsp. *Syriaca* GR-1-BBGK-15,5939 was classified in a group together with *S. montana* L., *S. romana*, *S. euboea*, and *S. scardica*, which are also found in Greece ([52]; Figure 1). The *Sideritis* specimen annotated with an asterisk in Figure 2 is denoted as *S. syriaca* in the GenBank database; however, a re-evaluation is warranted for this accession. Considering its collection locality (Greece: Mt. Dirfys; [6]) and given that *S. syriaca* has never been cultivated in Evia Island, it is suggested that the appropriate identification of this specimen should be *S. euboea*, which is the local endemic mountain tea species of Mt. Dirphys. This recommendation is substantiated by the restricted geographical distribution of *S. syriaca* subsp. *syriaca* on the island of Crete as a local single-island endemic plant [52]. Similar issues of misidentification or inaccurate taxonomy of specimens related to the focal taxon are extensively addressed in [17] for the case of the ITS2 molecular marker.

The Basic Local Alignment Search Tool (BLAST) results indicated the highest similarity of the new DNA sequences generated for *S. syriaca* subsp. *syriaca* GR-1-BBGK-15,5939 to the sequences retrieved from the GenBank. Since there was no previous information on sequence data for any *Sideritis* taxon for the cpDNA regions *pet*B/*pet*D, *rpo*C1, *psb*K-*psb*I, and *atp*F/*atp*H, the BLAST showed only other closely related taxa. The *trn*L/*trn*F showed the highest similarity to *S. scardica* (99.81%). The *rbc*L molecular marker of the present study was identical (100%) to *S. syriaca* subsp. *syriaca* and to another 19 specimens, corresponding to the Balkan endemics *S. raeseri* subsp. *raeseri* and *S. scardica* and the local Greek endemic *S. euboea*, all originating from Greece; in addition, a high similarity was shown to the *S. euboea* and *S. hyssopifolia*, which are wild-growing in France, Italy, Sicilia, Spain, and Switzerland [52]. It is worth mentioning that the molecular marker *rbc*L exhibited identical sequences across a notable array of *Sideritis* taxa, such as *S. perfoliata* L. subsp. *perfoliata*, *S. scardica*, *S. syriaca* subsp. *syriaca*, *S. clandestina* (Bory and Chaub.) Hayek subsp. *Clandestina*, and *S. euboea*, provided that all the available sequences for this molecular marker pertained to merely eight taxa. Although *psb*A/*trn*H is excluded from the phylogenetic analysis herein, its utilization is rather anticipated to yield inconclusive outcomes in terms of species differentiation. This assumption comes in contrast with previous studies [13] suggesting that the *mat*K and *psb*A/*trn*H could serve as potential single-region barcodes for Lamiaceae species. Additional phylogenetic analysis for another 94 specimens of the genus *Sideritis* from Greece showed identical sequences of three specimens of *S. syriaca* subsp. *Syriaca*; however, the rest of these specimens presented controversial matching and nucleotide differences to the assigned *Sideritis* species and subspecies (Appendix A). Recently, a new *Sideritis* species has been described from Bulgaria, namely, *S. elica* Aneva, Zhelev and Bonchev, which was clearly distinct from *S. scardica* based on morphological and molecular data [16]; in that case, the *trn*H-*psbA* molecular marker showed a 6.8% polymorphism [16]. Previous studies have also investigated the phylogeny of selected *Sideritis* taxa using only one nuclear molecular marker [14,17,55], thus compromising the genetic information for comparison with the current investigation.

The computation herein suggests that the combination of two or more molecular markers provides the potential for more informative species differentiation. In this study, we furnished, for the first time, genetic characterization based on seven molecular markers, thus consolidating the identity and authenticating the studied Cretan local endemic germplasm of *S. syriaca* subsp. *syriaca*, which is threatened with extinction [25].

### 3.2. Fertilization Effects Supporting the Sustainable Exploitation of S. syriaca *subsp.* syriaca

Plant species become extinct at a rate that is two to three orders higher than the expected natural one [56]. Overexploitation and habitat destruction or alteration currently threaten over 15,000 plant taxa with extinction [57], with a large part of their wild-growing populations currently depleted [58]. For species with increasing demand, adaptation to agricultural environments stands out as a sustainable conservation option. Successful cultivation, however, depends on the development of cultivation practices, a major part of which is the fertilization scheme. The present field study reports the primary stages of establishing *S. syriaca* subsp. *syriaca* (endemic to Crete, Greece) into systematic cultivation and evaluates the fertilization scheme that is optimal for improving plant growth as well as in terms of critical features of herbal material quality. In this perspective, emphasis was placed on the use of fertilizers having a low environmental footprint, which is of vital importance in the respective MAP market area [59].

As compared to controls, INM and ChF-sa improved plant growth (Figure 5A). With INM, this effect was more prominent under foliar application (Figure 5A). This increase in biomass accumulation was not associated with variation in tissue relative water content (Figure 5B). In this way, the efficiency of processing moisture content reduction through drying [31] was not affected by the fertilization scheme. The partitioning to the inflorescences was not significantly different among the three growth-promoting fertilization treatments (INM-fa, INM-sa, ChF-sa; Figure 5C). Considering yield, the current findings suggest that INM (especially through foliar application) and ChF-sa ought to be employed in *S. syriaca* subsp. *syriaca* cultivation.

Recent research on the pilot cultivation of Cretan endemic species in northern Greece (including *S. syriaca* subsp. *syriaca*) has also revealed noteworthy yield improvements with the implementation of comparable fertilization methods [60,61]. However, such enhanced yield was accompanied by distinct concentration patterns of macro- and micro-nutrients during the harvest stage. A balanced quantity of essential nutrients inside the plant is a vital factor influencing crop product quality and yield. Nutrient interactions are measured in terms of growth response and alteration in nutrients concentration. In this regard, the nutrient interactions are either positive (synergistic), negative (antagonistic), or neutral (no interaction) [62]. Specifically, our previous study has highlighted that conventional inorganic fertilization through foliar application (ChF-fa) may lead to elevated levels of metallic micro-nutrients at the harvest stage, specifically Cu and Zn [63]. Conversely, the same treatment, when applied to the soil, has been shown to result in a significant increase in Fe and Mn [63]. Nitrogen metabolites and N have been recently revealed to perform as vital growth regulators that may interplay with endogenous plant growth regulators, such as polyamines, in stress signaling procedures [64]. Interestingly, N concentration is reported to show minimal variation across different treatments compared to the control; whereas, P concentration may rise with foliar application using conventional inorganic fertilizers and K can exhibit positive responses with the integrated nutrient management (INM) fertilization approach through foliar application [63].

The dissimilarities between the present study’s findings in Crete and the aforementioned pilot case of cultivation in northern Greece could be attributed to the respective differences in the properties of the initial soil used as a medium, especially in terms of soil fertility. The soil in the present study (Crete) had elevated concentrations of soil-available macro- and micro-nutrients, exceeding their sufficiency levels [63]. In contrast, the soil utilized in our previously published study in northern Greece lacked sufficient soil-available potassium, falling below critical sufficiency levels, while the soil phosphorus content was only marginally sufficient [61]. The above indicates that any fertilization scheme suggestion in the future should be carefully adjusted with the initial amounts of soil available nutrients, taking also into consideration the complexity of soil–soil solution interactions that may interfere with the uptake process by plant roots.

Regarding the nutrient analysis of the inflorescences, the major part of *Sideritis herba* [24], it was found that all treatments tended to increase the macro-nutrient content, with the exception of P. The higher N content was observed in INM-sa; whereas, INM-fa and ChF-sa showed the highest *p* values. The latter treatments tended also to have higher K and Ca content (although not significantly different than the control), suggesting, overall, that ChF-sa and INM-fa may favor the accumulation of macro-nutrients in the inflorescences of *S. syriaca* subsp. *syriaca*. Regarding inflorescence micro-nutrient content, it seems that no significant enrichment of inflorescences can be referred to in terms of Fe, Zn, and B contents. No relative information was found in the literature to compare and evaluate the macro- and micro-nutrient contents in the inflorescences of *S. syriaca* subsp. *syriaca*. Thus, further work is needed for a greater period of time and under different experimental conditions to evaluate the partitioning of nutrients to inflorescences and the response of *S. syriaca* subsp. *syriaca* to fertilization.

Leaf green color intensity is an important visually perceived quality index throughout the supply chain [31]. In this study, leaf coloration was evaluated for the first time by three devices (SPAD meter, DA meter, Chroma Meter), indicating a different potential in discriminating the minor fertilization treatment effects (Figure 2, and Appendix A). At the full flowering stage, for instance, no difference among treatments was apparent when examining leaf SPAD value (Figure 2), L value (a measure of lightness; Appendix A), or b value (a measure of blue to yellow range intensity; Appendix A). Although little differences were noted in the I_AD_ value (Appendix A) and a value (a measure of green to red range intensity; Appendix A) among treatments, plants receiving fertilization did not differ from control plants. Therefore, it is concluded that the fertilization scheme did not affect leaf color aspects in the *S. syriaca* subsp. *syriaca*. Other than color, visual inspection in herbal material grading includes leaf form. As compared to control plants, fertilization increased both circularity and solidity (Figure 4B,D). Differences in circularity are translated to variation in lobing and serration while differences in solidity are translated to deep lobes or marked petiole morphology [65]. In this regard, trained personnel or automated identification protocols are expected to be capable of visually discriminating *S. syriaca* subsp. *syriaca* herbal material that has received fertilization during cultivation based on shape indicators since color differences were either absent or minor.

The beneficial effects of antioxidant intake are increasingly recognized by consumers [66]. Moreover, the antioxidant content of natural products (such as *Sideritis herba*) is becoming more prominent as an index of herbal material quality [34]. In this investigation, three critical antioxidants (carotenoids, flavonoids, phenols) were quantified and comparatively evaluated for the first time. MPE-sa was associated with reduced content in all three metabolites (Figure 6B–D), thus consistently downgrading quality. Fertilization mostly promoted carotenoid and total phenolic contents; whereas, it decreased flavonoid content (Figure 6B–D). The fertilization scheme stimulating antioxidant content was metabolite-specific (Figure 6B–D). Considering all three antioxidants together, ChF-fa was the most stimulatory scheme. Contrary to our results, INM has been previously associated with enhanced antioxidant compound content in other taxa [45,46].

Although the present study focused only on one growth cycle in Crete, it offered for the first time reference data for growth/yield aspects and several herbal quality aspects in *S. syriaca* subsp. *syriaca*. When the primary target is yield, INM (especially through foliar application) and ChF-sa ought to be employed. When antioxidant compound content is the primary goal, ChF-fa is required while MPE-sa ought to be avoided in *S. syriaca* subsp. *syriaca* cultivation.

## 4. Materials and Methods

### 4.1. Plant Material

Several botanical expeditions were conducted in the mountain ranges of Crete in the frame of the research project “Conservation and sustainable utilization of rare-threatened endemic plants of Crete for the development of new products with innovative precision fertilization” (acronym: PRECISE-M, Τ1ΕΔΚ-05380) with the aim to locate wild-growing populations of *S. syriaca* subsp. *syriaca* (Figure 8). Seed collections for ex situ conservation were authorized through a special permit of the Institute of Plant Breeding and Phytogenetic Resources, Hellenic Agricultural Organization, Demeter (Permit 82336/879 of 18 May 2019 and 26895/1527, 21 April 2021), obtained yearly by the Greek Ministry of Environment and Energy. The collected seeds and voucher samples were taxonomically identified and, consequently, unique IPEN (International Plant Exchange Network) accession numbers were assigned to each of them (GR-1-BBGK-08,4544; GR-1-BBGK-13,5743; GR-1-BBGK-13,5738; GR-1-BBGK-13,5907; GR-1-BBGK-14,5798; GR-1-BBGK-15,5939; GR-1-BBGK-17,6029; GR-1-BBGK-19,1100) according to the protocols of the Balkan Botanic Garden of Krousia, Institute of Plant Breeding and Genetic Resources, Hellenic Agricultural Organization, Demeter. An adequate number of plants was raised ex situ for the field experiment from seedlings and/or cuttings from mother plants of GR-1-BBGK-15,5939 with the aid of a specific propagation protocol [14]. All plant materials employed in the experimental procedure were transplanted in 2 L plastic pots by the company AFI GLAVAKI KE SIA OE Tree & Plant Nurseries, Aridea, PELLAS, GR-58400, Greece.

### 4.2. DNA Barcoding and Molecular Analysis

Molecular analyses were conducted using fresh juvenile leaves of *S. syriaca* subsp. *syriaca* GR-1-BBGK-15,5939. The methodology employed for DNA extraction, amplification, and subsequent DNA sequence analysis adhered to the procedures outlined by [67]. The oligonucleotide primers (5′ to 3′) (InVitrogen Inc., Paisley, Scotland, UK) used for the PCR amplification of *S. syriaca* subsp. *syriaca* were the following: ACTCGCACACACTCCCTTTCC and GCTTTTATGGAAGCTTTAACAAT for *atp*F/*atp*H [68], TTGACYCGTTTTTATAGTTTAC and AATTTAGCYCTTAATACAGG for *pet*B/*pet*D [68], CGCGCATGGTGGATTCACAATCC and GTTATGCATGAACGTAATGCTC for *trn*H/*psb*A [69], TTAGCCTTTGTTTGGCAAG and AGAGTTTGAGAGTAAGCAT [70] for *psb*K-*psb*I, ATGTCACCACAAACAGAGACTAAAGC and CTTCTGCTACAAATAAGAATCGATCTC for *rbc*L [71], GGCAAAGAGGGAAGATTTCG and CCATAAGCATATCTTGAGTTGG for *rpo*C1 [72], ATTTGAACTGGTGACACGAG and CGAAATCGGTAGACGCTACG for *trn*L/*trn*F [73]. The annealing temperature for each pair of primers was 54 to 60 °C, depending on sequence temperature estimation.

The alignment of each generated sequence was completed by employing the Basic Local Alignment Search Tool (BLAST), comparing them with existing sequences available in the GenBank. DNA sequences corresponding to the above-mentioned molecular markers for other *Sideritis* taxa were retrieved from GenBank and were subsequently aligned for each marker using Mega11 software [74]. A unified phylogenetic tree was constructed using the neighbor-joining statistical method and the maximum composite likelihood substitution model. To ensure accessibility and transparency, all newly generated sequences were submitted to the GenBank obtaining the accession numbers OR909054–OR909060. The evaluation of evolutionary divergence between sequences for each molecular marker involved the calculation of pair-wise distances, conducted within the Mega11 software [74].

### 4.3. Field Experiment and Experimental Design

The field experiment was established at the beginning of March 2021 within a 20 × 25 m fenced area of the campus of the Hellenic Mediterranean University, Heraklion, Crete (35°19′ N, 25°6′ E) at a low altitude (60 m above sea level). The planting distance between individuals was 40 cm and 80 cm between separate rows 20 m long, arranged in an east–west direction and including as “guard plants” other local Cretan endemics, i.e., plant individuals of *Origanum dictamnus* L., *Origanum microphyllum* (Benth.) Vogel, *Carlina diae* (Rech.f.) Meusel and A.Kástner, and *Verbascum arcturus* L.

The experimental design incorporated completely randomized blocks of ten plants of *S. syriaca* subsp. *syriaca* per block and three blocks per treatment, randomly arranged in different rows. To provide the same starting point for all experimental plants, all individuals were trimmed at 5 cm above ground level at the end of April. At the end of May (i.e., 30 d after trimming), six fertilization treatments were introduced weekly until the final harvest. An automatic irrigation system was employed with 2 L h^−1^ adjustable drippers to supply water to the plants three times per week. Pest and disease control was not deemed necessary during experimentation; however, the removal of weeds was regularly required and manually performed. The final harvest was carried out at the end of June 2021. The soil properties of the pilot field are described in our previous studies [63,75].

### 4.4. Fertilization Treatments

The pilot cultivation of *S. syriaca* subsp. *syriaca* followed weekly fertilization treatments using water (control), conventional inorganic (ChF), biostimulant, and INM in liquid or soluble granule fertilizers administered with foliar and soil applications (Appendix A). The foliar applications were performed using a 5 L plastic handheld sprayer (low pressure) until apparent wetness and the soil applications were manually performed (100 mL of nutrient solution per plant). The INM and biostimulants were supplied with four special fertilizers from THEOFRASTOS company, Industrial Area of Korinthos, GR-20100 Korinthos, Greece (Appendix A). These polysaccharide-based semi-organic fertilizers are made from high-quality organic edible raw materials and plant extracts, resulting in a full supplement of plants with amino acids, vitamins, sugars, and macro-microelements [63,75]. The INM polysaccharides are aimed to contribute to the formation of stable soil aggregates formed in connection with active soil microorganisms; INM polysaccharides, when foliar-sprayed, are expected to be rapidly absorbed by the plants. The conventional inorganic fertilizers were all in soluble powder or granule form, except the liquid fertilizer for micro-nutrients (Plex Mix, AGRI.FE.M. LTD Fertilizers, Aspropyrgos, Greece). Two foliar fertilization treatments (INM and ChF), three soil application treatments (INM, ChF, and a biostimulator), and a control for the pilot cultivation of *S. syriaca* subsp. *syriaca* in the field were employed in the experimentation (Appendix A).

### 4.5. Plant Measurements

All plant and leaf level measurements were regularly conducted until 25 May 2021, ending with the completion of flowering. Regarding leaf measurements, fully expanded leaves grown under direct light were sampled within 15 min. When this was not feasible, samples were placed in vials, were flash-frozen in liquid nitrogen, and were transferred to a freezer (−80 °C) for storage. All replicate leaves were sampled from separate plant individuals.

#### 4.5.1. Non-Invasive Evaluation of Leaf Coloration at Three Growth Stages

Since leaf coloration depends on the fertilization scheme affecting both photosynthetic capacity and visually perceived quality, this variable was assessed by employing three methods:(i)Leaf SPAD value, approximating chlorophyll content, was determined by using a SPAD-502 (Konica Minolta Corp., Solna, Sweden);(ii)Index of absorbance difference (I_AD_) accurately evaluated fruit ripeness since it is closely associated with outer mesocarp chlorophyll content [76]; I_AD_ was computed as the difference between the absorbance values at 670 and 720 nm, near the chlorophyll absorbance peak [76]. The potential of I_AD_ in reflecting respective differences in leaves had not been previously evaluated. However, the estimation of their chlorophyll index could provide an indication of their maturity state. Thus, the I_AD_ index could help in providing the developmental stage of a measured leaf and could be worthwhile in estimating the steps of the leaf product chain and the potential marketing cycle of the product. In this investigation, I_AD_ was determined in leaves by using the DA meter (tr DA Meter, T.R. Turoni, Italy);(iii)Leaf color was quantified by using a Chroma Meter (Model CR-400, Minolta Corp., Osaka, Japan); CIE L*a*b* coordinates were recorded using D65 illuminants and a 10° Standard Observer as a reference system: L* (a measure of lightness, ranging from 0 (black) to 100 (white)), a* (a measure of intensity in the green to red range, where negative values refer to green and positive to red), and b* (a measure of representing intensity in the blue to the yellow range, where negative values refer to blue and positive to yellow).

These measurements were in situ conducted in attached leaves of intact plant individuals at three different growth stages (vegetative, early flowering, and full flowering). Three points were recorded per replicate leaf and were further averaged. Three replicate leaves were assessed per treatment.

#### 4.5.2. Non-Invasive Evaluation of Photosynthetic Performance in Growth Stages

As a valid indicator of leaf photosynthetic performance, the ratio of variable to maximum chlorophyll fluorescence (F_v_/F_m_) was assessed. Measurements were performed by using a chlorophyll fluorometer (OS-30P, Op-tiSciences, Hudson, NH, USA). Prior to evaluation, leaves were dark-adapted (≥20 min) by employing leaf clips. F_v_/F_m_ was assessed by applying a saturated photosynthetic photon flux density of 3000 µmol m^−2^ s^−1^.

These measurements were in situ conducted on the attached leaves of intact plant individuals and at vegetative, early flowering, and full flowering stages. Three points were recorded per replicate leaf and were further averaged. Three replicate leaves were assessed per treatment.

#### 4.5.3. Leaf Shape Indicators

A morphometric analysis was performed by analyzing leaf form. Leaf shape traits were derived from images acquired by a digital camera (Sony DSC-W830, Sony Corporation, Tokyo, Japan) under non-reflective glass from a distance of 0.5 m, employing a copy stand. Using specialized software (ImageJ2; Wayne Rasband/NIH, Bethesda, MD, USA), leaf lamina outlines were processed to estimate the following four (dimensionless) metrics of leaf form: (a) aspect ratio [(major axis)/(minor axis); axes of the best-fitted ellipse], (b) circularity [(4π × area)/(perimeter)^2^], (c) roundness [(4 × area)/[4π × (major axis)^2^]], and (d) solidity [(area)/(convex area)] [77]. Each metric was used to quantify a distinct feature of leaf shape. Aspect ratio and roundness are affected by the length-to-width ratio while circularity and solidity depend on serration and lobing [65]. The aspect ratio ranges from 1 (circle) to a value without upper bound (infinitely narrow). Roundness ranges from 0 (infinitely narrow) to 1 (circle). Circularity ranges from 0 (infinitely narrow) to 1 (circle). Solidity ranges from 0 to 1, being inversely related to boundary irregularities. Solidity is sensitive to leaves with deep lobes or a distinct petiole and can be employed to detect leaves lacking such structures [65]. Solidity, unlike circularity, is not greatly influenced by serrations and minor lobing [65]. In total, 30 leaves (5 per plant × 6 plant individuals) were analyzed per treatment.

#### 4.5.4. Plant Growth and Biomass Partitioning to Generative Organs

Plant growth and biomass partitioning to generative organs were determined. Above-ground and inflorescence (fresh and dry) biomasses were determined (±0.01 g; MXX-412; Denver Instruments, Bohemia, NY, USA). For measuring dry weight, samples were placed in a forced-air-drying oven for 72 h at 80 °C. Six replicate plants were evaluated per treatment.

#### 4.5.5. Leaf Chlorophyll and Carotenoid Contents

Chlorophyll content is important for leaf coloration and photosynthetic performance and carotenoids represent key non enzymatic antioxidants. The effect of growth conditions on chlorophyll and carotenoid contents was therefore assessed. Following fine chopping, leaf portions (0.5 g) were homogenized with the addition of 10 mL of 80% acetone. This primary acetone extract was then filtered and the filtered extract was diluted by adding 2 mL of 80% acetone per mL of extract. Since chlorophyll is light-sensitive, extraction took place in a dark room. The obtained extract was subjected to reading on a spectrophotometer (Mapada UV-1800; Shanghai. Mapada Instruments Co., Ltd., Shanghai, China). Total chlorophyll and carotenoid contents were calculated [78]. Three leaves were assessed per treatment. For each replicate, four samples (collected from different plant individuals) were pooled and the assay was performed twice.

#### 4.5.6. Leaf Total Phenolic and Total Flavonoid Contents

Phenols and flavonoids are critical non enzymatic antioxidants. Leaf samples (0.1 g) were extracted with 1 mL of 80% aqueous methanol in an ultrasonic bath (10 min) and were then centrifuged (15,000× *g* for 10 min). The contents of total phenolics and total flavonoids were determined by using the Folin–Ciocalteu assay and the aluminum chloride colorimetric assay, respectively. The absorbance against the prepared reagent blank was determined using a microplate reader (Infinite 200 PRO, TECAN, Zürich, Switzerland). For total phenolic content, gallic acid was used as the standard reference and gallic acid equivalent (GAE) was expressed as mg per g fresh mass. For total flavonoid content, rutin was used as the standard reference and rutin equivalent (RUE) was expressed as mg per g fresh mass. Three leaves were assessed per treatment. For each replicate, four samples (collected from different plant individuals) were pooled and the assay was performed twice.

#### 4.5.7. Soluble Sugar Content in Leaves

The carbohydrate status is related to photosynthetic activity. The effect of the fertilization scheme on carbohydrate status was, therefore, assessed. Leaf samples (0.1 g) were incubated with 1 mL of deionized water in a water bath (100 °C for 30 min). The homogenate was centrifuged (15,000× *g* for 15 min) at room temperature (25 °C). Then, 0.1 mL of the solution was mixed with anthranone ethyl acetate and sulphuric acid. Soluble sugar content was assayed in the supernatant according to [79] by measuring the absorbance at 630 nm using a spectrophotometer (Mapada UV-1800; Mapada Instruments Co., Ltd., Shanghai, China). These measurements were conducted on three replicates per treatment. For each replicate, four samples (collected from different plant individuals) were pooled and the assay was performed twice.

#### 4.5.8. Leaf and Inflorescence Nutrient Analysis

To assess the role of the fertilization scheme in nutrient uptake by plants, leaf and inflorescence nutrient analysis was conducted. The samples were washed with distilled water, dried at 70 °C, weighed, ground, and then analyzed for total N by the Kjeldahl method [80]. In addition, sub-samples were ash-burned at 500 °C for at least 4 h; the ash was dissolved in 2 M HCl; filtered; and P, K, Ca, Mg, Cu, Zn, Fe, Mn, and B were determined in the filtrate by flame photometry, atomic absorption spectrometry, and UV-vis spectrometry, depending on the element. Nutrient content was expressed on a dry weight basis. Three replicates were evaluated per treatment. For each replicate, four samples (collected from different plant individuals) were pooled and the assay was performed twice.

### 4.6. Statistical Analysis

Data analyses of the fertilization experiment were carried out using the SPSS version 22 (IBM SPSS Statistics). Data were tested for homogeneity of variances (Duncan’s test). One-way analysis of variance (ANOVA) was conducted on data and the least significant differences (LSD) test was applied to detect significant differences among means (*p* = 0.05).

## 5. Conclusions

In this study, consolidated genetic fingerprinting based on seven molecular markers (four markers used for the first time in members of genus *Sideritis*) and DNA barcoding using two molecular markers was generated for the first time for *S. syriaca* subsp. *syriaca*, a threatened local endemic plant of Crete (Greece) with depleting wild-growing populations due to over-harvesting from the natural environment, thus consolidating its molecular identity and enabling the traceability of related commercial products. Due to limited cultivation/fertilization data, the optimal fertilization scheme for plant growth/yield and herbal material quality was investigated in *S. syriaca* subsp. *syriaca* in a field study to facilitate its local cultivation in Crete as a valuable MAP with strong medicinal value approved by the European Medicines Agency. Five fertilization schemes were examined, including conventional inorganic fertilizers (foliar/soil application), INM (foliar/soil application), and INM with biostimulants (soil application). Leaf color was not altered by the fertilization scheme but leaf shape was indeed affected. In this way, the visually perceived quality was influenced by fertilization. INM, especially by foliar application, and conventional inorganic fertilizers by soil application improved yield and did not alter tissue water content or biomass partitioning to generative organs. Conventional inorganic fertilizers (foliar application) were the best treatment for enhanced antioxidant compound content; whereas, INM with biostimulants was the worst one. In conclusion, different fertilization choices may be employed when targeting either yield or antioxidant compound accumulation in *S. syriaca* subsp. *Syriaca*. For the former, INM (foliar application) and conventional inorganic fertilization (soil application) were found to be suitable; whereas, for the latter, the best scheme found was conventional inorganic fertilization (foliar application). The data furnished herein are aimed to further facilitate the sustainable exploitation of this valuable medicinal-aromatic plant still suffering from over-harvesting, directly wild-growing populations threatened with extinction.

## Figures and Tables

**Figure 1 ijms-25-01891-f001:**
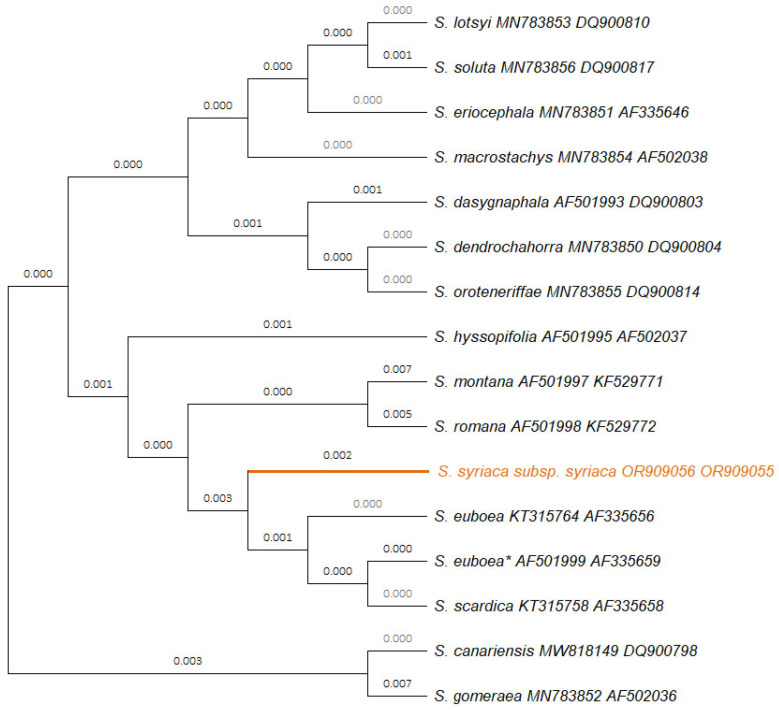
Phylogenetic tree illustrating the relationships among 15 *Sideritis* species based on the molecular plastid markers *rbc*L and *trn*L/*trn*F. The tree was constructed using the neighbor-joining method and maximum composite likelihood substitution model. Accession numbers of DNA sequences obtained in this study are indicated next to taxon names. The clade marked in orange represents the *Sideritis syriaca* subsp. *syriaca* GR-1-BBGK-15,5939 studied herein. The taxon marked with an asterisk (*) is characterized as *S. syriaca* in the database; however, based on its original collection area (Mt. Dirphys in Evia, Greece), it should be identified as the unique local endemic mountain tea plant of this mountain, namely, *S. euboea* and not as *S. syriaca* subsp. *Syriaca*, which is a single-island endemic of Crete Island, Greece [52].

**Figure 2 ijms-25-01891-f002:**
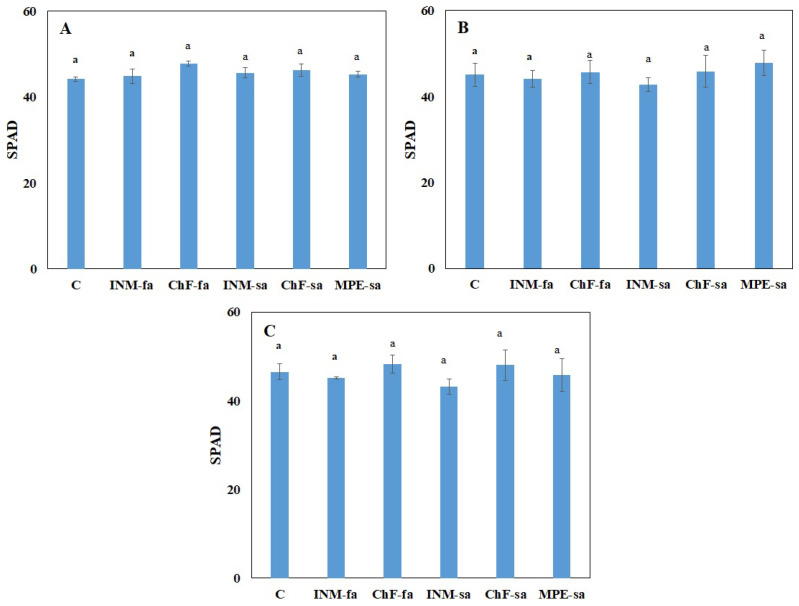
Effect of fertilization schemes applied through foliar or soil on the leaf SPAD value of *Sideritis syriaca* subsp. *syriaca* at the vegetative (**A**), early flowering (**B**), and full flowering (**C**) stages. C: Control (water); INM-fa: Integrated nutrient management (INM) by foliar application; ChF-fa: Conventional inorganic fertilization (ChF) by foliar application; INM-sa: INM by soil application; ChF-sa: ChF by soil application; MPE-sa: Mixture of plant extracts as biostimulant by soil application (THEOMASS). Columns represent the means of three replicates ± SEM. Within each plot, different letters indicate significant differences among means.

**Figure 3 ijms-25-01891-f003:**
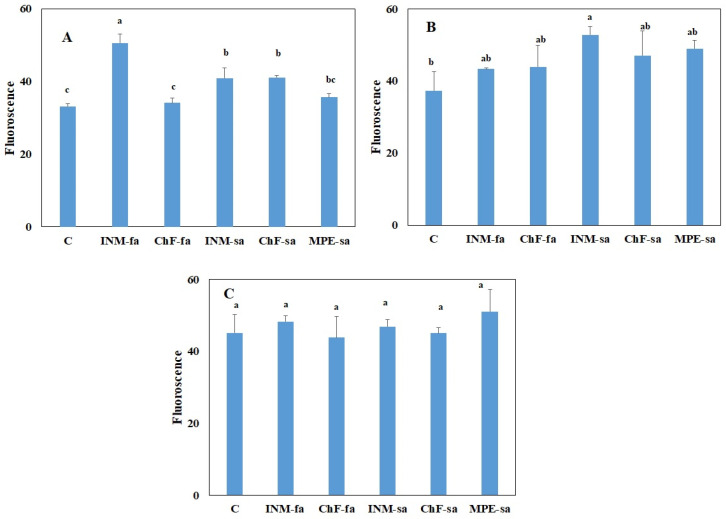
Effect of fertilization schemes applied through foliar or soil on the chlorophyll fluorescence value of *Sideritis syriaca* subsp. *syriaca* at the vegetative (**A**), early flowering (**B**), and full flowering (**C**) stages. C: Control (water); INM-fa: Integrated nutrient management (INM) by foliar application; ChF-fa: Conventional inorganic fertilization (ChF) by foliar application; INM-sa: INM by soil application; ChF-sa: ChF by soil application; MPE-sa: Mixture of plant extracts as biostimulant by soil application (THEOMASS). Columns represent the means of three replicates ± SEM. Within each plot, different letters indicate significant differences among means.

**Figure 4 ijms-25-01891-f004:**
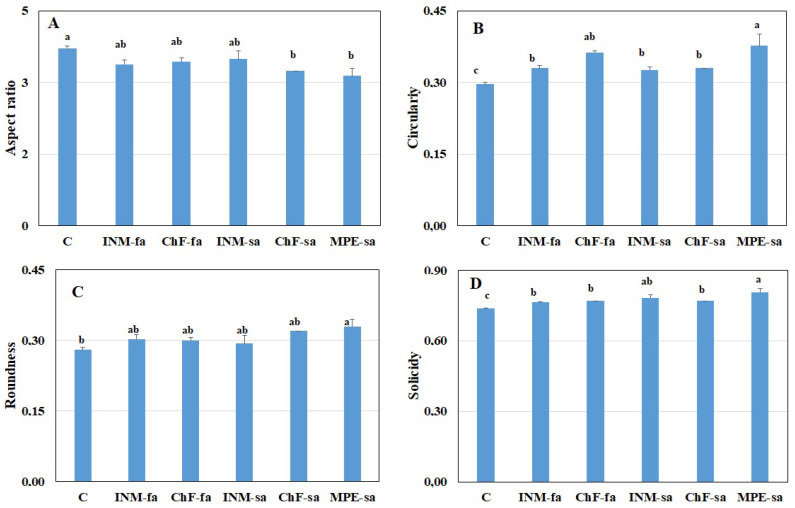
Effect of fertilization schemes applied through foliar or soil on the four leaf shape factors (**A**, Aspect ratio; **B**, Circularity; **C**, Roundness; **D**, Solidity) of *Sideritis syriaca* subsp. *syriaca*. C: Control (water); INM-fa: Integrated nutrient management (INM) by foliar application; ChF-fa: Conventional inorganic fertilization (ChF) by foliar application; INM-sa: INM by soil application; ChF-sa: ChF by soil application; MPE-sa: Mixture of plant extracts as biostimulant by soil application (THEOMASS). Columns represent the means of six replicates ± SEM. Within each plot, different letters indicate significant differences among means.

**Figure 5 ijms-25-01891-f005:**
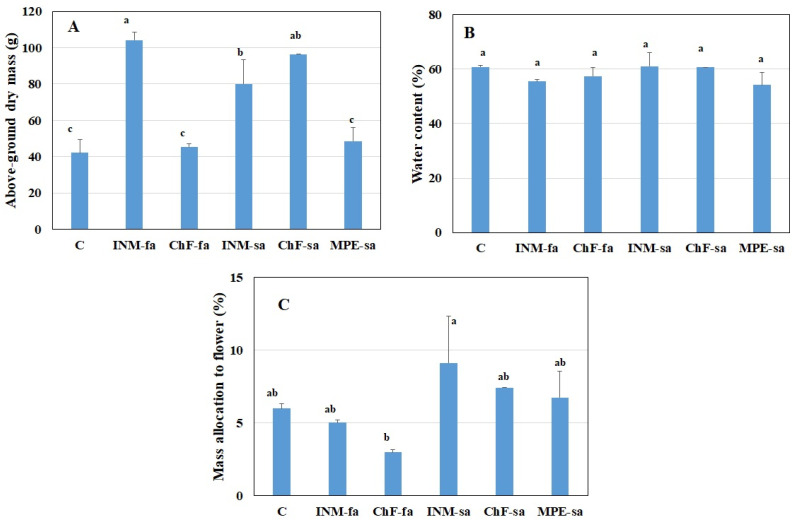
Effect of fertilization schemes applied through foliar or soil on the above-ground dry weight (**A**), water content (**B**), and dry weight partitioning to inflorescences (**C**) of *Sideritis syriaca* subsp. *syriaca*. C: Control (water); INM-fa: Integrated nutrient management (INM) by foliar application; ChF-fa: Conventional inorganic fertilization (ChF) by foliar application; INM-sa: INM by soil application; ChF-sa: ChF by soil application; MPE-sa: Mixture of plant extracts as biostimulant by soil application (THEOMASS). Columns represent the means of six replicates ± SEM. Within each plot, different letters indicate significant differences among means.

**Figure 6 ijms-25-01891-f006:**
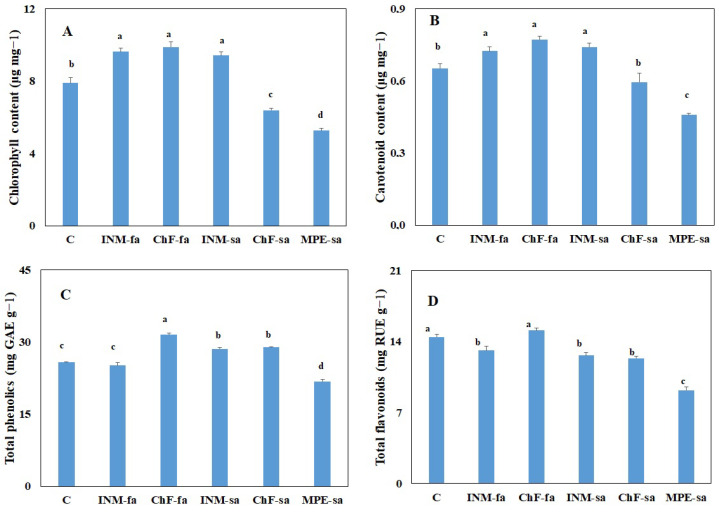
Effect of fertilization schemes applied through foliar or soil on the leaf chlorophyll (**A**), carotenoid (**B**), total phenol (**C**), and total flavonoid (**D**) content of *Sideritis syriaca* subsp. *syriaca*. C: Control (water); INM-fa: Integrated nutrient management (INM) by foliar application; ChF-fa: Conventional inorganic fertilization (ChF) by foliar application; INM-sa: INM by soil application; ChF-sa: ChF by soil application; MPE-sa: Mixture of plant extracts as biostimulant by soil application (THEOMASS). Columns represent the means of three replicates ± SEM. Within each plot, different letters indicate significant differences among means.

**Figure 7 ijms-25-01891-f007:**
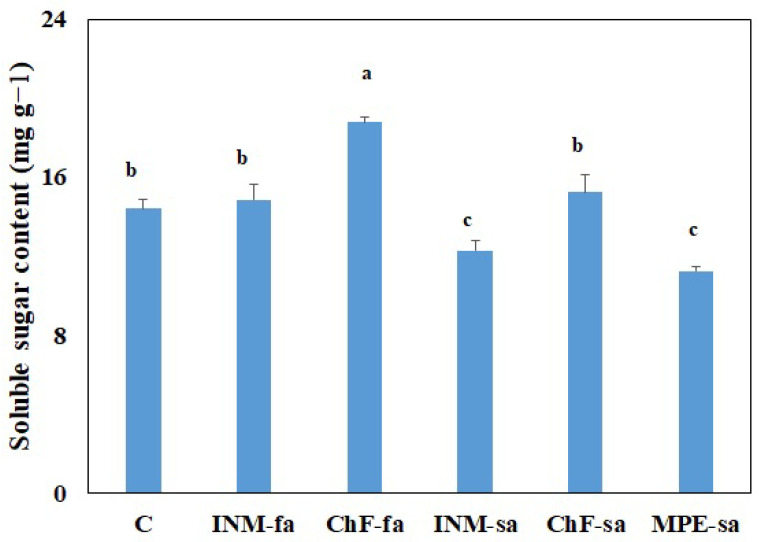
Effect of fertilization schemes applied through foliar or soil on the leaf-soluble sugar content of *Sideritis syriaca* subsp. *syriaca*. C: Control (water); INM-fa: Integrated nutrient management (INM) by foliar application; ChF-fa: Conventional inorganic fertilization (ChF) by foliar application; INM-sa: INM by soil application; ChF-sa: ChF by soil application; MPE-sa: Mixture of plant extracts as biostimulant by soil application (THEOMASS). Columns represent the means of three replicates ± SEM. Different letters indicate significant differences among means.

**Figure 8 ijms-25-01891-f008:**
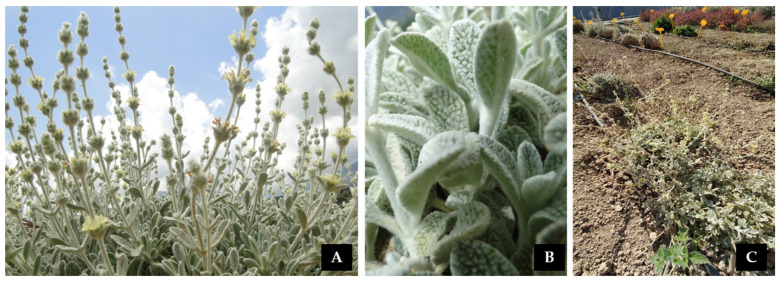
(**A**) *Sideritis syriaca* subsp. *syriaca* wild-growing individual at full flowering on rocky crevices of Sfakianes Madares in Lefka Ori, Chania, Crete (Greece); (**B**) Densely hairy leaf indumentum; and (**C**) Pilot field with *S. syriaca* subsp. *syriaca* experimental individuals cultivated at the premises of the Hellenic Mediterranean University, Heraklion, Crete (Greece).

**Table 1 ijms-25-01891-t001:** Effect of fertilization schemes applied through foliar or soil on the leaf essential macro-nutrient content of *Sideritis syriaca* subsp. *syriaca*.

Treatment	N	P	K	Ca	Mg
	(g kg^−1^)				
C	13.4 ± 0.4 c*	2.3 ± 0.1 b	18.9 ± 0.3 b	23.0 ± 1.4 a	3.2 ± 0.1 a
INM-fa	21.8 ± 1.7 ab	3.4 ± 0.0 a	18.7 ± 0.4 b	18.7 ± 0.5 a	2.7 ± 0.0 a
ChF-fa	18.3 ± 0.3 abc	3.5 ± 0.0 a	22.4 ± 0.5 a	22.9 ± 1.8 a	3.0 ± 0.1 a
INM-sa	22.8 ± 0.8 a	2.5 ± 0.1 b	18.7 ± 0.5 b	22.1 ± 1.7 a	2.9 ± 0.1 a
ChF-sa	12.8 ± 1.6 c	2.7 ± 0.0 b	20.4 ± 0.4 ab	23.3 ± 1.5 a	4.2 ± 0.5 a
MPE-sa	16.0 ± 1.5 bc	2.3 ± 0.2 b	19.1 ± 0.5 b	25.6 ± 1.0 a	3.2 ± 0.1 a
*p* F-test	0.021	<0.001	0.029	NS ^#^	NS

C: Control (water); INM-fa: Integrated nutrient management (INM) by foliar application; ChF-fa: Conventional inorganic fertilization (ChF) by foliar application; INM-sa: INM by soil application; ChF-sa: ChF by soil application; MPE-sa: Mixture of plant extracts as biostimulator by soil application (THEOMASS). Values represent the means of three replicates ± SE. * Means followed by different letters within the same column are statistically different using the LSD test at *p* ≤ 0.05. ^#^ NS: Non-significant.

**Table 2 ijms-25-01891-t002:** Effect of fertilization schemes applied through foliar or soil on the leaf essential micro-nutrient content of *Sideritis syriaca* subsp. *syriaca*.

Treatment	Cu	Zn	Fe	Mn	B
	(mg kg^−1^)				
C	14.9 ± 0.1 abc*	26.5 ± 1.4 a	1058 ± 89 cd	52.0 ± 1.6 c	27.5 ± 0.8 b
INM-fa	13.2 ± 0.4 c	23.8 ± 0.5 a	1057 ± 49 cd	62.8 ± 1.3 bc	19.7 ± 0.3 d
ChF-fa	14.5 ± 0.5 bc	27.5 ± 0.4 a	1277 ± 79 bc	83.0 ± 5.0 a	26.9 ± 0.3 bc
INM-sa	15.8 ± 0.3 ab	24.7 ± 0.6 a	1703 ± 111 a	68.4 ± 1.1 b	25.8 ± 0.3 bc
ChF-sa	15.5 ± 0.6 ab	23.3 ± 0.1 a	871 ± 19 d	37.0 ± 0.4 d	24.1 ± 0.8 c
MPE-sa	16.9 ± 0.2 a	26.8 ± 1.0 a	1578 ± 24 ab	54.0 ± 1.5 c	35.0 ± 0.6 a
*p* F-test	0.033	NS ^#^	0.003	<0.001	<0.001

C: Control (water); INM-fa: Integrated nutrient management (INM) by foliar application; ChF-fa: Conventional inorganic fertilization (ChF) by foliar application; INM-sa: INM by soil application; ChF-sa: ChF by soil application; MPE-sa: Mixture of plant extracts as biostimulant by soil application (THEOMASS). Values represent the means of three replicates ± SE. * Means followed by different letters within the same column are different using the LSD test at *p* ≤ 0.05. ^#^ NS: Non-significant.

**Table 3 ijms-25-01891-t003:** Effect of fertilization schemes applied through foliar or soil on the essential macro-nutrient content of *Sideritis syriaca* subsp. *syriaca* inflorescences.

Treatment	N	P	K	Ca	Mg
	(mg kg^−1^)				
C	8.65 ± 0.00 c*	1.31 ± 0.45 b	12.67 ± 2.22 ab	1.84 ± 0.19 bc	1.04 ± 0.20 ab
INM-fa	10.14 ± 0.36 bc	2.40 ± 0.38 a	13.95 ± 0.44 ab	2.75 ± 0.31 ab	1.28 ± 0.01 a
ChF-fa	10.28 ± 2.07 bc	1.72 ± 0.12 ab	9.19 ± 1.24 b	1.22 ± 0.01 c	0.61 ± 0.12 b
INM-sa	16.63 ± 1.68 a	1.26 ± 0.31 b	12.21 ± 1.16 ab	3.20 ± 0.69 ab	1.27 ± 0.29 a
ChF-sa	10.46 ± 0.00 bc	2.64 ± 0.00 a	16.84 ± 0.00 a	3.49 ± 0.00 a	1.40 ± 0.00 a
MPE-sa	13.38 ± 1.92 ab	1.61 ± 0.53 ab	12.52 ± 2.52 ab	2.56 ± 1.04 ab	1.36 ± 0.35 a
*p* F-test	0.014	0.075	0.081	0.083	NS ^#^

C: Control (water); INM-fa: Integrated nutrient management (INM) by foliar application; ChF-fa: Conventional inorganic fertilization (ChF) by foliar application; INM-sa: INM by soil application; ChF-sa: ChF by soil application; MPE-sa: Mixture of plant extracts as bioreactor by soil application (THEOMASS). Values represent the means of three replicates ± SEM. * Means followed by different letters within the same column are different using the LSD test at *p* ≤ 0.05. ^#^ NS: Non-significant.

**Table 4 ijms-25-01891-t004:** Effect of fertilization schemes applied through foliar or soil on the essential micro-nutrient content of *Sideritis syriaca* subsp. *syriaca* inflorescences.

Treatment	Β	Zn	Fe
	mg kg^−1^		
C	6.93 ± 0.12 a*	14.81 ± 3.20 ab	149.86 ± 43.79 ab
INM-fa	5.60 ± 0.20 b	15.32 ± 1.41 ab	201.35 ± 42.26 a
ChF-fa	6.60 ± 0.34 ab	10.78 ± 2.93 b	49.94 ± 23.79 b
INM-sa	6.82 ± 0.78 a	14.30 ± 1.20 ab	135.70 ± 38.84 ab
ChF-sa	5.49 ± 0.00 b	20.20 ± 1.00 a	157.32 ± 10.00 ab
MPE-sa	6.29 ± 0.18 ab	17.51 ± 4.47 ab	148.68 ± 56.23 ab
*p* F-test	0.068	NS ^#^	NS

C: Control (water); INM-fa: Integrated nutrient management (INM) by foliar application; ChF-fa: Conventional inorganic fertilization (ChF) by foliar application; INM-sa: INM by soil application; ChF-sa: ChF by soil application; MPE-sa: Mixture of plant extracts as bioreactor by soil application (THEOMASS). Values represent the means of three replicates ± SEM. * Means followed by different letters within the same column are different using the LSD test at *p* ≤ 0.05. ^#^ NS: Non-significant.

## Data Availability

All data supporting the results of this study are included in the article and its Appendix A. Original datasets are also available upon request All newly generated DNA sequences were submitted to the GenBank under the accession numbers OR909054-OR909060.

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
