# Peer review of "DNA Barcoding and Fertilization Strategies in Sideritis syriaca subsp. syriaca, a Local Endemic Plant of Crete with High Medicinal Value"

_ijms, 2024, doi:10.3390/ijms25031891_

Round 1

Reviewer 1 Report

Comments and Suggestions for Authors

I appreciate the well written manuscript entitled " DNA Barcoding and Fertilization Strategies in Sideritis syriaca subsp. syriaca, a Local Endemic Plant of Crete with High Medicinal Value” which applied DNA Barcoding for the genetic characterization of Sideritis syriaca  subsp. syriaca using seven molecular markers of cpDNA. Within the context of the work, it can be mentioned that the topic is interesting as well as I found it interesting that the search for information was done through an experimental design, a methodology that is very useful when it is required to verify different experimental parameters and their effect within an analytical process. The introduction presents sufficient preliminary information to understand the foundation and purpose of the scientific work developed by the authors. Likewise, an evaluation was made of probable plagiarism, but nothing was found that indicated it, so it is considered that this is a completely original work. It is worth mentioning that the experiments within the experimental design were adequately directed, the information presented on the experimental methodologies carried out is perfectly founded both in its proposal within the experimental determinations, as well as in the particular application. The statistical tools used show consistent results within the analysis carried out by the authors.

That being said, the manuscript has the potential to be accepted after minor revision. While the authors should address some issues before acceptance

Minor comments:

Lines 31-34: The presentation of the key findings of experimental results should be improved and data regarding the mainly measured indicators should be presented.

Lines 593: The authors should give detailed information about the composition of biostimulant used in this experiment.

The authors should make efforts to enhance the quality of the figures.

Figure 2A: Please align the letters of the statistics above the columns.

Kind Regards.

Comments on the Quality of English Language

Minor editing of English language required

Author Response

REVIEWER 1 - Comments and Suggestions for Authors

I appreciate the well written manuscript entitled "DNA Barcoding and Fertilization Strategies in Sideritis syriaca subsp. syriaca, a Local Endemic Plant of Crete with High Medicinal Value” which applied DNA Barcoding for the genetic characterization of Sideritis syriaca subsp. syriac using seven molecular markers of cpDNA. Within the context of the work, it can be mentioned that the topic is interesting as well as I found it interesting that the search for information was done through an experimental design, a methodology that is very useful when it is required to verify different experimental parameters and their effect within an analytical process. The introduction presents sufficient preliminary information to understand the foundation and purpose of the scientific work developed by the authors. Likewise, an evaluation was made of probable plagiarism, but nothing was found that indicated it, so it is considered that this is a completely original work. It is worth mentioning that the experiments within the experimental design were adequately directed, the information presented on the experimental methodologies carried out is perfectly founded both in its proposal within the experimental determinations, as well as in the particular application. The statistical tools used show consistent results within the analysis carried out by the authors.

That being said, the manuscript has the potential to be accepted after minor revision.

Authors’ response: We would like to thank the reviewer for his/her favorable comments suggesting our paper as suitable for publication after minor revision.

While the authors should address some issues before acceptance

Minor comments:

Lines 31-34: The presentation of the key findings of experimental results should be improved and data regarding the mainly measured indicators should be presented.

Authors’ response: The presentation of the key findings in the abstract has been improved as suggested by the reviewer (see track changes).

Lines 593: The authors should give detailed information about the composition of biostimulant used in this experiment.

Authors’ response: The composition of biostimulant has been analytically cited in previous publications (Paschalidis, K.; Fanourakis, D.; Tsaniklidis, G.; Tzanakakis, V.A.; Bilias, F.; Samara, E.; Kalogiannakis, K.; Debouba, F.J.; Ipsilantis, I.; Tsoktouridis, G.; et al. Pilot cultivation of the vulnerable cretan endemic Verbascum arcturus L. (Scrophulariaceae): Effect of fertilization on growth and quality features. Sustainability 2021, 13. https://doi.org/10.3390/su132414030; Fanourakis, D.; Paschalidis, K.; Tsaniklidis, G.; Tzanakakis, V.A.; Bilias, F.; Samara, E.; Liapaki, E.; Jouini, M.; Ipsilantis, I.; Maloupa, E.; et al. Pilot cultivation of the local endemic Cretan marjoram Origanum microphyllum (Benth.) Vogel (Lamiaceae): Effect of fertilizers on growth and herbal quality features. Agronomy 2022, 12. https://doi.org/10.3390/agronomy12010094). To avoid self-plagiarism and duplication, we have preferred to refer clearly to these publications (references 57,69) in the Material and Methods section (see track changes). However, the raw materials composed the biostimulant is secret and involves the “Know how” of this particular product. We have asked the company for more information and they told us that the raw materials derived from a mixture of natural plant extracts which contain NO algae.

The authors should make efforts to enhance the quality of the figures.

Authors’ response: The quality of the figures has now been improved, perhaps it was only a pdf effect (the word file contains original photos in high resolution).

Figure 2A: Please align the letters of the statistics above the columns.

Authors’ response: The letters of the statistics have been aligned above the columns as suggested by the reviewer (see track changes).

Reviewer 2 Report

Comments and Suggestions for Authors In this study, the authors provide important insights into the genetic makeup of Sideritis syriaca subsp. syriaca, a locally threatened plant with medicinal value in Crete. Utilizing DNA Barcoding, they analyze seven cpDNA markers, shedding light on the species' genetic intricacies. Simultaneously, they assess five fertilization strategies, including chemical fertilizers, integrated nutrient management, and biostimulants. Their findings not only reveal the genetic nuances of S. syriaca subsp. syriaca but also provide insights into optimal fertilization practices for sustainable cultivation, emphasizing the delicate balance between genetic characterization and plant physiology. However, there is still room for improvement before it can be accepted. 

Minor comments    1. Line 48: Specify the molecular markers used for molecular barcoding in other MAPs to enhance context and comparability.   2. Line 50-54: Elaborate on the specific challenges within the Lamiaceae family, especially regarding the complex taxonomical issues in the genus Sideritis.   3. Line 56-58: Provide a brief summary or key findings of the first DNA study on Sideritis taxa using RAPDs for better context. 4. Line 60-63: Clarify the significance of the datasets deposited in GenBank for Macaronesian Sideritis taxa and Greek Sideritis taxa, emphasizing their contribution to the broader understanding of genetic relatedness.   5. Line 71-74: Elaborate on the ecological niche and significance of S. syriaca L. subsp. syriaca, considering its high-altitude habitat and flowering patterns.   6. Line 181-184: Discuss the implications of the identical DNA sequence for rbcL among S. syriaca subsp. syriaca, S. raeseri subsp. raeseri, S. euboea, and S. scardica. What does this reveal about their genetic relatedness?   7. Line 187-189: Elaborate on the nucleotide differences among 90 more Sideritis taxa and the challenges they pose, discussing potential reasons or consequences.   8. Line 192-196: Provide insight into the significance of the BLAST results, especially the similarities with other plant species, and discuss any potential implications.   9. Line 198-204: Clarify the meaning and significance of IAD values, especially in relation to leaf color. Provide a concise interpretation of the findings.   10. Line 270-273: Elaborate on the potential implications of the observed differences in leaf chlorophyll content due to different fertilization treatments. Discuss how this might affect the overall plant health and performance.   11. Line 277-279: Provide insights into the ecological significance of variations in carotenoid content, especially considering their role as non-enzymatic antioxidants.   12. Line 281-283: Discuss the importance of total phenolic content and its variations among different fertilization treatments, considering the potential health benefits associated with phenolic compounds.   13. Line 285-286: Elaborate on the observed differences in flavonoid content, explaining their potential implications for the plant's response to different fertilization regimes.   14. Line 296-297: Discuss the significance of variations in leaf soluble sugar content among different treatments. Address how these differences may impact the plant's energy balance or other physiological processes.   15. Line 306-308: Discuss the implications of varying nutrient concentrations in leaves, particularly the observed high N concentration under integrated nutrient management fertilization (INM-sa).   16. Line 309-311: Consider providing a brief discussion on the observed trends in K and P concentrations and their potential impact on plant growth and development.   17. Line 312: Mention the lack of significant differences in Ca and Mg concentrations and briefly discuss the potential reasons for these results.   18. Line 314-320: Discuss the significance of micronutrient concentrations, particularly the observed effects on Fe and Mn concentrations under different fertilization treatments. Comments on the Quality of English Language

See my comments above 

Author Response

REVIEWER 2 - Comments and Suggestions for Authors

In this study, the authors provide important insights into the genetic makeup of Sideritis syriaca subsp. syriaca, a locally threatened plant with medicinal value in Crete. Utilizing DNA Barcoding, they analyze seven cpDNA markers, shedding light on the species' genetic intricacies. Simultaneously, they assess five fertilization strategies, including chemical fertilizers, integrated nutrient management, and biostimulants. Their findings not only reveal the genetic nuances of S. syriaca subsp. syriaca but also provide insights into optimal fertilization practices for sustainable cultivation, emphasizing the delicate balance between genetic characterization and plant physiology. However, there is still room for improvement before it can be accepted.

Authors’ response: We would like to thank the reviewer for his/her kind words.

Minor comments

  1. Line 48: Specify the molecular markers used for molecular barcoding in other MAPs to enhance context and comparability.

Authors’ response: The information has been added as suggested by the reviewer (see track changes). The molecular markers used for molecular barcoding in other MAPs and especially Sideritis spp. are reviewed in terms of context and for comparability in Lines 59-73.

  1. Line 50-54: Elaborate on the specific challenges within the Lamiaceae family, especially regarding the complex taxonomical issues in the genus Sideritis.

Authors’ response: As suggested by the reviewer, new information has been added regarding specific challenges and taxonomical issues in wild-growing and ex situ cultivated Sideritis spp. (see track changes).

  1. Line 56-58: Provide a brief summary or key findings of the first DNA study on Sideritis taxa using RAPDs for better context.

Authors’ response: As suggested by the reviewer, new information has been added regarding the first DNA study on Sideritis taxa using RAPDs for better context (see track changes).

  1. Line 60-63: Clarify the significance of the datasets deposited in GenBank for Macaronesian Sideritis taxa and Greek Sideritis taxa, emphasizing their contribution to the broader understanding of genetic relatedness.

Authors’ response: As suggested by the reviewer, the significance of these datasets has been outlined in the respective passages (see track changes).

  1. Line 71-74: Elaborate on the ecological niche and significance of S. syriaca L. subsp. syriaca, considering its high-altitude habitat and flowering patterns.

Authors’ response: The ecological significance considering its high-altitude habitat and flowering pattern has been elaborated in the revised version of the manuscript (see track changes).

  1. Line 181-184: Discuss the implications of the identical DNA sequence for rbcL among S. syriaca subsp. syriaca, S. raeseri subsp. raeseri, S. euboea, and S. scardica. What does this reveal about their genetic relatedness?

Authors’ response: In this part of the results section we pointed out the problems of mislabeling or wrongly identified Sideritis samples at species level. The last sentence of this paragraph we state the following: “Surprisingly, there were nucleotide differences among 90 more Sideritis taxa originated from Greece with results shown to be highly problematic (Supplementary Material 3.B)”

We cannot write more things because we do not have more details of the samples deposited in the GenBank and second the article has not been published yet. We assume that will be a reaction from the people who deposited all of these problematic DNA sequences.

  1. Line 187-189: Elaborate on the nucleotide differences among 90 more Sideritis taxa and the challenges they pose, discussing potential reasons or consequences.

Authors’ response: As mentioned previously, we reported the problems of these samples and DNA sequences for the 90 samples of Sideritis and these people must check their data again prior publication. In my opinion they have mislabeled all their samples during experimental process (possibly immature students’ work during their training).

  1. Line 192-196: Provide insight into the significance of the BLAST results, especially the similarities with other plant species, and discuss any potential implications.

Authors’ response: Regarding comments 6, 7 and 8 (Lines 181-184, 187-189 and 192-196), we would like to mention that these implications are already discussed in lines 424-463 of the revised manuscript. However, to address the reviewer’s comments we have further amended these texts (see track changes).

  1. Line 198-204: Clarify the meaning and significance of IAD values, especially in relation to leaf color. Provide a concise interpretation of the findings.

Authors’ response: The meaning and significance of IAD values has been accordingly given in detail in Materials and Methods (Lines 680-683 of the revised manuscript).

  1. Line 270-273: Elaborate on the potential implications of the observed differences in leaf chlorophyll content due to different fertilization treatments. Discuss how this might affect the overall plant health and performance.

Authors’ response: As advised, the potential implications of the observed differences in leaf chlorophyll content due to different fertilization treatments have been better presented explaining how they might affect the overall plant health and performance, further citing the respective review article “Chlorophyll as a measure of plant health” (see track changes).

  1. Line 277-279: Provide insights into the ecological significance of variations in carotenoid content, especially considering their role as non-enzymatic antioxidants.

Authors’ response: Following the reviewer’s suggestion, insights into the carotenoids’ role as non-enzymatic antioxidants have been provided in the revised version of the manuscript, further citing the respective review article “Prooxidant Activity of Polyphenols, Flavonoids, Anthocyanins and Carotenoids” (see track changes).

  1. Line 281-283: Discuss the importance of total phenolic content and its variations among different fertilization treatments, considering the potential health benefits associated with phenolic compounds.

Authors’ response: Following the reviewer’s suggestion, insights into the phenolics’ role have been provided in the revised version of the manuscript considering the potential plant benefits, further citing the respective review article “Prooxidant Activity of Polyphenols, Flavonoids, Anthocyanins and Carotenoids” (see track changes).

  1. Line 285-286: Elaborate on the observed differences in flavonoid content, explaining their potential implications for the plant's response to different fertilization regimes.

Authors’ response: As suggested, insights into the flavonoids’ role considering the potential plant benefits have been provided in the revised version of the manuscript, further citing the respective review article “Prooxidant Activity of Polyphenols, Flavonoids, Anthocyanins and Carotenoids” (see track changes).

  1. Line 296-297: Discuss the significance of variations in leaf soluble sugar content among different treatments. Address how these differences may impact the plant's energy balance or other physiological processes.

Authors’ response: Following the reviewer’s suggestion, insights into the leaf soluble sugar significance in plant’s energy balance have been provided in the revised version of the manuscript, further citing the respective review article “The Role of Sugars in Plant Responses to Stress and Their Regulatory Function” (see track changes).

  1. Line 306-308: Discuss the implications of varying nutrient concentrations in leaves, particularly the observed high N concentration under integrated nutrient management fertilization (INM-sa).

Authors’ response: As advised, insights into the implication of nitrogen concentration as a key factor in plant response to different fertilization schedules have been provided in the revised version of the manuscript, further citing the respective review article “The Interplay among Polyamines and Nitrogen in Plant Stress Responses” (see track changes).

  1. Line 309-311: Consider providing a brief discussion on the observed trends in K and P concentrations and their potential impact on plant growth and development.

Authors’ response: A brief discussion on nutrient interactions and the potential impact on plant growth and development has been provided in the revised version of the manuscript, further citing the respective review article “Nutrient interactions in crop plants” (see track changes).

  1. Line 312: Mention the lack of significant differences in Ca and Mg concentrations and briefly discuss the potential reasons for these results.

Authors’ response: A brief discussion on nutrient interactions and the potential impact on plant growth and development has been provided in the revised version of the manuscript, further citing the respective review article “Nutrient interactions in crop plants” (see track changes).

  1. Line 314-320: Discuss the significance of micronutrient concentrations, particularly the observed effects on Fe and Mn concentrations under different fertilization treatments.

Authors’ response: A brief discussion on nutrient interactions and the potential impact on plant growth and development has been provided in the revised version of the manuscript, further citing the respective review article “Nutrient interactions in crop plants” (see track changes).

Reviewer 3 Report

Comments and Suggestions for Authors

Dear Authors,

The subject of the study is interesting and topical, with high scientific and practical importance.

The introduction is presented correctly, in accordance with the subject. Numerous scientific articles, in concordance to the topic of the study, were consulted.

Methodology of the study was clearly presented, and appropriate to the proposed objectives.

The obtained results are important and have been analyzed and interpreted correctly, in accordance with the current methodology.

The discussions are appropriate, in the context of the results, and was conducted compared to other studies in the field.

The scientific literature, to which the reporting was made, is recent and representative in the field.

Some minor suggestions and corrections were made in the article.

The following aspects are brought to the attention of the authors.

1.

Suggestion

Page 2, row 78

If the link of the consulted bibliographic source is cited in the References chapter, can be replaced in the text with the bibliographic source number "[]"

Similarly

Page 2, rows 82 – 83.

The authors of the study can decide better on this aspect.

2.

Italic Font style for species name

e.g.

page 21, rows 878 – 879

Sideritis raeseri

Sideritis syriaca

3.

Capital letter for the author, in the name of the species

e.g.

page 22, row 946

“Bornm.”

Author Response

REVIEWER 3

The subject of the study is interesting and topical, with high scientific and practical importance. The introduction is presented correctly, in accordance with the subject. Numerous scientific articles, in concordance to the topic of the study, were consulted.

Methodology of the study was clearly presented, and appropriate to the proposed objectives.

The obtained results are important and have been analyzed and interpreted correctly, in accordance with the current methodology. The discussions are appropriate, in the context of the results, and was conducted compared to other studies in the field. The scientific literature, to which the reporting was made, is recent and representative in the field.

Authors’ response: We would like to thank the reviewer for his/her favorable comments and for suggesting our paper for publication.

Some minor suggestions and corrections were made in the article.

The following aspects are brought to the attention of the authors.

  1. Suggestion Page 2, row 78: If the link of the consulted bibliographic source is cited in the References chapter, can be replaced in the text with the bibliographic source number "[]"

Authors’ response: Corrected as suggested.

Similarly Page 2, rows 82 – 83; The authors of the study can decide better on this aspect.

Authors’ response: Corrected as suggested.

  1. ItalicFont style for species name e.g. page 21, rows 878 – 879 “Sideritis raeseri

Sideritis syriaca

Authors’ response: Corrected as suggested.

  1. Capital letter for the author, in the name of the species e.g.

page 22, row 946 “Bornm.”

Authors’ response: Corrected as suggested.

Reviewer 4 Report

Comments and Suggestions for Authors

Comments and Suggestions for Authors

In the present manuscript the rwsults of genetic characterization of Sideritis syriaca  subsp. syriaca; a threatened, local Cretan endemic plant, were presented. The study was made on the base of DNA Barcoding with seven molecular markers of cpDNA. As result of the study new DNA sequence  datasets for the plastid regions of petB/petD, rpoC1, psbK-psbI and atpF/atpH have been deposited in the GenBank for S. syriaca subsp. syriaca while the molecular markers rbcL, trnL/trnF, psbA/trnH were compared to those of another 15 Sideritis species retrieved from the GenBank, constructing a phylogenetic tree to show their genetic relatedness.

The manuscript is composed according to Author guidlines of “International Journal of Molecular Sciences”. The methods used in the research are appropriate and sufficient to achieve the objectives of the study. The results are presented well and supported by tables and figures of good quality, and by statistical analysis.

The following recommendations can be made:

Results

The titles of subdivisions (2.2-2.8) need some changes, as follows:

All these sections to be united in one section: 2.2. Effect of fertilization scheme on the leaf structure, color and active compounds accumulation. Sections 2.2.-2.8. will become subsections:

2.2.1. Effect of fertilization scheme on the leaf color

 Fertilization scheme exerted minor effects on leaf colour. At all three growth stages, ….

2.2.2. Effect of fertilization scheme on the photosynthesis

Fertilization scheme induced limited effects on leaf photosynthetic performance. At three growth stages, the effect of…

2.2.3. Effect of fertilization scheme on the leaf shape

Fertilization induced distinct leaf shape profiles. To determine the effect of fertilization regime on leaf form, …

2.2.4. Effect of foliar fertilization ( INM-fa, ChF-fa) on the biomass accumulation

NM and ChF-sa stimulated plant growth, without affecting biomass allocation to generative  organs. INM with either application method (i.e., INM-fa or INM-sa) and ChF-sa improved  biomass accumulation as …

2.2.5. Effect of fertilization scheme on chlorophyll content

Fertilization regime affected leaf chlorophyll content. Plant receiving foliar fertilization (i.e., INM-fa, ChF-fa) or INM-sa had …

2.2.6. Effect of fertilization scheme on sugar and leaf antioxidant compound content

Fertilization regime affected leaf antioxidant compound content. Carotenoids, flavonoids, and phenols are …

Fertilization regime affected leaf soluble sugar content. Plants receiving ChF-sa had higher leaf soluble sugar content ….

and “2.9. Leaf and inflorescence mineral analysis” will become number 2.3

The passage on lines 159-163 is more suitable for Discussion part: “No data was available in the GenBank … [18].”

The legend in the table titles should be moved to the footnote. Example: In Table 1, “Effect of fertilization regime by different (root/foliar) application methods on the content of essential macronutrients in the leaves of Sideritis syriaca subsp. syriaca” in the title, and “C: Control (water); INM-fa…” – in the footnote. Likewise in the other tables.

Materials and Methods

In line 613: “…until 25 May 25 2021” something is missing, maybe it should be “..25 May and 25 …?2021”

In conclusion, this manuscript is recommended for publication in “International Journal of Molecular Sciences”.

Author Response

REVIEWER 4

In the present manuscript the results of genetic characterization of Sideritis syriaca subsp. syriaca; a threatened, local Cretan endemic plant, were presented. The study was made on the base of DNA Barcoding with seven molecular markers of cpDNA. As result of the study new DNA sequence datasets for the plastid regions of petB/petD, rpoC1, psbK-psbI and atpF/atpH have been deposited in the GenBank for S. syriaca subsp. Syriaca while the molecular markers rbcL, trnL/trnF, psbA/trnH were compared to those of another 15 Sideritis species retrieved from the GenBank, constructing a phylogenetic tree to show their genetic relatedness.

The manuscript is composed according to Author guidelines of “International Journal of Molecular Sciences”. The methods used in the research are appropriate and sufficient to achieve the objectives of the study. The results are presented well and supported by tables and figures of good quality, and by statistical analysis.

Authors’ response: We would like to thank the reviewer for his/her favorable comments.

The following recommendations can be made:

Results

The titles of subdivisions (2.2-2.8) need some changes, as follows:

All these sections to be united in one section: 2.2. Effect of fertilization scheme on the leaf structure, color and active compounds accumulation. Sections 2.2.-2.8. will become subsections:

2.2.1. Effect of fertilization scheme on the leaf color

Fertilization scheme exerted minor effects on leaf colour. At all three growth stages, ….

2.2.2. Effect of fertilization scheme on the photosynthesis

Fertilization scheme induced limited effects on leaf photosynthetic performance. At three growth stages, the effect of…

2.2.3. Effect of fertilization scheme on the leaf shape

Fertilization induced distinct leaf shape profiles. To determine the effect of fertilization regime on leaf form, …

2.2.4. Effect of foliar fertilization ( INM-fa, ChF-fa) on the biomass accumulation

NM and ChF-sa stimulated plant growth, without affecting biomass allocation to generative organs. INM with either application method (i.e., INM-fa or INM-sa) and ChF-sa improved biomass accumulation as …

2.2.5. Effect of fertilization scheme on chlorophyll content

Fertilization regime affected leaf chlorophyll content. Plant receiving foliar fertilization (i.e., INM-fa, ChF-fa) or INM-sa had …

2.2.6. Effect of fertilization scheme on sugar and leaf antioxidant compound content

Fertilization regime affected leaf antioxidant compound content. Carotenoids, flavonoids, and phenols are …

Fertilization regime affected leaf soluble sugar content. Plants receiving ChF-sa had higher leaf soluble sugar content ….

and “2.9. Leaf and inflorescence mineral analysis” will become number 2.3

Authors’ response: In the results section, the titles of subdivisions 2.2-2.8 are intentionally structured to depict in a clear way the most significant findings of this study. This widely used strategy was chosen to facilitate comprehension, readership, and results’ interpretation. In this regard, we would prefer to keep the results’ titles unchanged unless the academic editor advises us differently.

The passage on lines 159-163 is more suitable for Discussion part: “No data was available in the GenBank … [18].”

Authors’ response: We agree with the reviewer. Therefore, this passage has been transferred to the beginning of the Discussion section (see track changes).

The legend in the table titles should be moved to the footnote. Example: In Table 1, “Effect of fertilization regime by different (root/foliar) application methods on the content of essential macronutrients in the leaves of Sideritis syriaca subsp. syriaca” in the title, and “C: Control (water); INM-fa…” – in the footnote. Likewise in the other tables.

Authors’ response: We agree with the reviewer. The explanations in legends of all tables have been transferred to the footnotes of the respective tables in the revised version of the manuscript (see track changes).

Materials and Methods

In line 613: “…until 25 May 25 2021” something is missing, maybe it should be “..25 May and 25 …?2021”

Authors’ response: Corrected as suggested (see track changes).

In conclusion, this manuscript is recommended for publication in “International Journal of Molecular Sciences”.

Authors’ response: We would like to thank the reviewer for suggesting our paper as suitable for publication.